# In situ structure and assembly of the multidrug efflux pump AcrAB-TolC

Xiaodong Shi[1,2,8], Muyuan Chen[1,8], Zhili Yu[1], James M. Bell[1,3], Hans Wang[1], Isaac Forrester[4], Heather Villarreal[4], Joanita Jakana[4], Dijun Du[5,7], Ben F. Luisi [5], Steven J. Ludtke [1] & Zhao Wang [1,6]

Multidrug efflux pumps actively expel a wide range of toxic substrates from the cell and play a major role in intrinsic and acquired drug resistance. In Gram-negative bacteria, these pumps form tripartite assemblies that span the cell envelope. However, the in situ structure and assembly mechanism of multidrug efflux pumps remain unknown. Here we report the in situ structure of the *Escherichia coli* AcrAB-TolC multidrug efflux pump obtained by electron cryo-tomography and subtomogram averaging. The fully assembled efflux pump is observed in a closed state under conditions of antibiotic challenge and in an open state in the presence of AcrB inhibitor. We also observe intermediate AcrAB complexes without TolC and discover that AcrA contacts the peptidoglycan layer of the periplasm. Our data point to a sequential assembly process in living bacteria, beginning with formation of the AcrAB subcomplex and suggest domains to target with efflux pump inhibitors.

[1] Verna and Marrs McLean Department of Biochemistry and Molecular Biology, Baylor College of Medicine, Houston, TX 77030, USA. [2] Jiangsu Province Key Laboratory of Anesthesiology and Jiangsu Province Key Laboratory of Anesthesia and Analgesia Application, Xuzhou Medical University, Xuzhou, Jiangsu 221004, China. [3] Graduate Program in Quantitative and Computational Biosciences, Baylor College of Medicine, Houston, TX 77030, USA. [4] CryoEM Core at Baylor College of Medicine, Houston, TX 77030, USA. [5] Department of Biochemistry, University of Cambridge, Cambridge CB21GA, UK. [6] Department of Molecular and Cellular Biology, Baylor College of Medicine, Houston, Texas 77030, USA. [7] Present address: School of Life Science and Technology, ShanghaiTech University, Shanghai 201210, China. [8] These authors contributed equally: Xiaodong Shi, Muyuan Chen. Correspondence and requests for materials should be addressed to Z.W. (email: zhaow@bcm.edu)

With the increasing use of antibiotics, multidrug resistance in pathogenic bacteria has become a public health crisis. The capability of numerous bacterial species to survive in the presence of antibiotics and toxic compounds is partially conferred by the activity of energy-dependent efflux pumps[1,2]. In Gram-negative bacteria, these pumps are multicomponent assemblies that span the cell envelope and are driven by a primary or a secondary transport component located in the inner membrane[3]. AcrAB–TolC is one of the tripartite pumps that are constitutively expressed in *Escherichia coli* (*E. coli*)[4,5]. As the main multidrug efflux machinery, AcrAB–TolC is comprised of the outer membrane protein TolC, the periplasmic adaptor protein AcrA, and the inner membrane transporter AcrB from the resistance-nodulation-cell division (RND) superfamily[3]. The AcrAB–TolC efflux pump transports diverse compounds, conferring resistance to a broad spectrum of antibiotics[6]. Structural studies of this pump have been limited to individual components by X-ray crystallography[7–14] or fully assembled pumps by cryo-electron microscopy (cryo-EM) single-particle analysis[15–19]. These approaches revealed structures in vitro, but the in situ structure of this pump remains unknown. Due to the dynamic nature of the three components and their low binding affinities, it is particularly challenging to capture the intermediate states of the AcrAB–TolC pump in vitro, and there is only limited information about the assembly mechanism of the pump in living cells. Here, we visualize the in situ structure of *E. coli* AcrAB–TolC efflux pump by employing cellular electron cryo-tomography (cryo-ET) and subtomogram averaging. Our results reveal in situ structures of the fully assembled pump and its intermediate assembly state and suggest an assembly mechanism for tripartite efflux pumps in Gram-negative bacteria.

## Results

**Visualization of AcrAB–TolC pump in *E. coli* cell envelope**. To enrich AcrAB–TolC pumps in situ, we overexpressed AcrA, AcrB, and TolC in BL21 (DE3) cells at a level at which the cells can still replicate and grow (Supplementary Fig. 1). Then we imaged cells with Cryo-ET under antibiotic treatment that promotes pump assembly[20]. Three-dimensional tomographic reconstructions revealed detailed structures of the Gram-negative bacterial envelope, with abundant channel-like densities spanning the cell envelope (Fig. 1 and Supplementary Movie 1). These densities are rarely observed in wild-type *E. coli* cells (Supplementary Fig. 2), implying that they correspond to AcrAB–TolC pumps. In addition, the distance between the outer membrane and the inner membrane stays constant at the sites where the AcrAB–TolC pumps occur, suggesting that the periplasm may be pinched by these assemblies.

**In situ structures of the fully assembled AcrAB–TolC complex**. In order to determine the in situ structure of AcrAB–TolC pump, we extracted particles of the cell envelope spanning densities and performed subtomogram averaging. From 1321 subtomograms of the AcrAB–TolC pump with C3 symmetry, we achieved a reconstruction at ~15 Å resolution (gold standard FSC, see "Methods" section) (Supplementary Figs. 3, 4). The averaged map resembles the EM structure of the AcrAB–TolC pump, with a length of ~33 nm (Fig. 2a–c). The in situ arrangement of each component of the pump matches the previous cryo-EM studies[15–17]. The overall architecture of the fully assembled pump clearly indicates a 3:6:3 ratio for TolC: AcrA: AcrB in situ (Fig. 2b, c), which agrees with our previous cryo-EM structures[16,19]. Notably, the density occupancy in the TolC region is considerably lower than the rest of the structure, suggesting that TolC may be absent in a subset of the particles (Supplementary Fig. 5a). We

then performed focused classification with a soft spherical mask on the upper part of the pump (Supplementary Fig. 5b). This classification yielded two maps, one showing the full pump with a visible TolC density, and the other one containing only the AcrAB subassembly of the pump and the density of the cell membranes (Supplementary Figs. 5c, 6).

In the fully assembled pump, the inner chamber shows a clear constriction (Fig. 2a, c), which has only been observed in the apo-form of an AcrA-AcrB crosslinked pump in previous in vitro experiments[19]. Therefore, we concluded that the map corresponded to a closed state and it fitted our higher resolution published structure (PDB: 5V5S) well. Despite the presence of the antibiotic, we did not capture the transporting state where a continuous conduit is formed between TolC and AcrA. However, our drug resistance test showed that the strain overexpressing the AcrAB–TolC pumps had a much higher minimum inhibitory concentration (MIC) than the wild-type strain (Supplementary Table 1), indicating that the AcrAB–TolC pumps are functional. Thus, there must be pumps that are in a transporting state in the bacteria, in order to produce the antibiotic resistance. To validate that the AcrAB–TolC pumps can open in our system, we treated the cells with MBX3132, an inhibitor of AcrB, that is known to lock AcrB of the pump in vitro[19]. According to our MIC results, both the wild-type strain and the AcrAB–TolC pump overexpressing strain became hyper-resistant to puromycin in the presence of MBX3132 (Supplementary Table 1). Using the same data collection and processing protocol, we captured the open state pump that had a continuous conduit through TolC and AcrA. The structure has a length of 32 nm, shorter than that of closed state (Fig. 2d and Supplementary Fig. 7), consistent with our in vitro results obtained by cryo-EM[19]. The data indicate that contraction along the long axis is part of active transport in vivo. Taken together, our data suggest that the opening of the pump and efflux of antibiotics is likely a transient process, with the majority of the complexes observed at any given time in a closed state, rather than an active state.

Compared to single-particle cryo-EM, cellular Cryo-ET provides the capability of visualizing the interactions of a protein complex with its surroundings. In the tomograms of bacteria under antibiotic pressure, we found that the density of the PG layer is situated just above the top of AcrA when the cell envelope densities are overlaid with the averaged map of the full pump (Fig. 2e). These observations suggest that the PG is contacted both by the coiled-coils of TolC and the α-hairpin loop of AcrA, instead of the equatorial domain of TolC as previously proposed[21]. These interactions are validated by the mask-free subtomogram average of the same particles (Supplementary Fig. 8). In addition, our in vivo crosslinking experiment mapped the binding sites of both AcrA and TolC with PG (Supplementary Figs. 9, 10), further verifying our observation that both AcrA and TolC interact with PG in the complex.

**In situ structure of the AcrAB subcomplex**. As mentioned above, the three-dimension classification of subtomograms revealed ~38% of the particles within the dataset do not have TolC located in the outer membrane (see "Methods" section). The averaged density map of these particles represents a bipartite AcrAB subcomplex (Fig. 3a–c), in which the quaternary organization of AcrA and AcrB is similar to a proposed assembly model based on the crystal structure of the recombinant heavy-metal efflux pump CusBA[22]. In the structure of AcrAB subcomplex, six protruding densities of AcrA in the averaged map indicate a 6:3 ratio between AcrA and AcrB, the same as the fully assembled pump (Supplementary Fig. 5c). The interior of AcrA viewed in a cross-section through the averaged map of the subcomplex differs

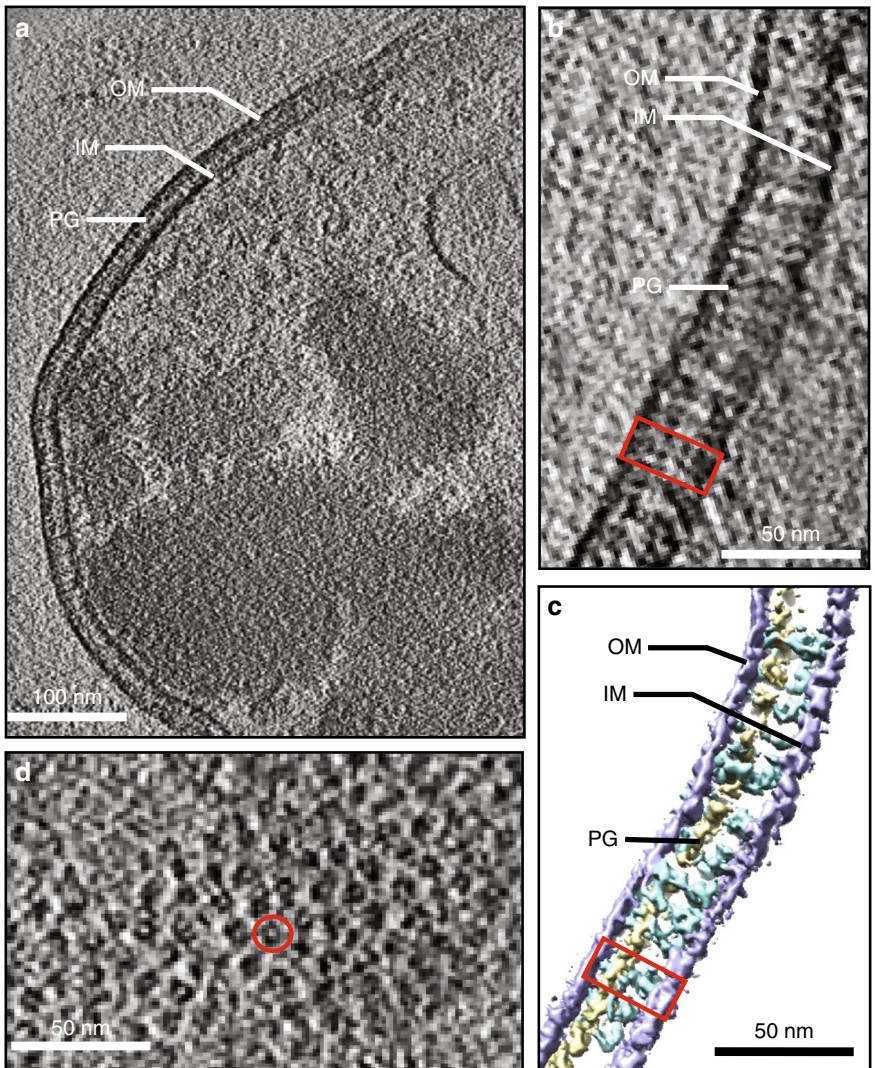

**Fig. 1** Visualizing the AcrAB–TolC efflux pump in the *E. coli* cell envelope. **a** A single slice from a tomogram of *E. coli*. The condensed materials shown inside of the cell are inclusion bodies resulting from membrane protein overexpression. **b** Zoomed inside view of the cell envelope containing the AcrAB–TolC pump which is indicated by the red rectangle. **c** Corresponding three-dimensional annotation of **b** showing the outer membrane (OM; blue), the inner membrane (IM; blue), peptidoglycan (PG; yellow), and the AcrAB–TolC pump (cyan). **d** Top view of the cell envelope containing the AcrAB–TolC pumps which appear as ring-shaped densities (indicated by the red circle)

from that of the fully assembled pump, with the chamber inside AcrA being smaller (Fig. 3c) and the AcrA hairpin domains not forming a complete ring structure (Supplementary Fig. 5c). The helical hairpin region of AcrA repacks to form an alpha-helical barrel in the transition from the apo to the ligand-bound states of the AcrAB–TolC assembly, and the reorganization of the AcrA hexamer is likely to be a critical step for the opening of the TolC channel to form the active tripartite pump complex[19].

In the unmasked average, the tip of AcrA density merges into the density of PG layer, while the space between the PG and the outer membrane is empty (Fig. 3d). The PG layer possibly serves as an anchor to hold the AcrA hexamer in the periplasm to maintain the stability of AcrAB subcomplex in the envelope or to help the bipartite AcrAB subcomplex recruit TolC. Our finding that the AcrAB subcomplex exists as a stable entity in cells is supported by previous experiments both in vivo and in vitro[20,23,24]. We did not observe any complex directly between AcrB and TolC, which is consistent with prior structural studies[15,16,19]. Likewise, AcrA–TolC complexes were not detected in cells either.

## Discussion
In conclusion, we suggest that the pump assembly process follows a sequential order. Based on the tomography results and data from in vivo interactions, it is likely that AcrB and AcrA can associate to form a bipartite complex. The contact between the α-hairpin domain of AcrA and PG helps to position and maintain the stability of the AcrAB complex in the cell envelope, and may permit the subcomplex to walk along the layer until it encounters TolC (Fig. 4). In the presence of antibiotics, the AcrAB subcomplex changes its conformation to recruit TolC, which remains closed in the outer membrane to keep the periplasm isolated from the extracellular environment. Notably, the in situ fully assembled closed state pump showed a constriction in between TolC and AcrA. Next, the pump briefly adopts an open conformation accompanied with a contraction to promote the expulsion of the substrate through the chamber and closes immediately after the drug molecule is expelled.

In this study, we captured the fully assembled pumps on *E. coli* membranes exhibiting a closed state in the presence of antibiotics and an open state in the presence of the AcrB inhibitor. In

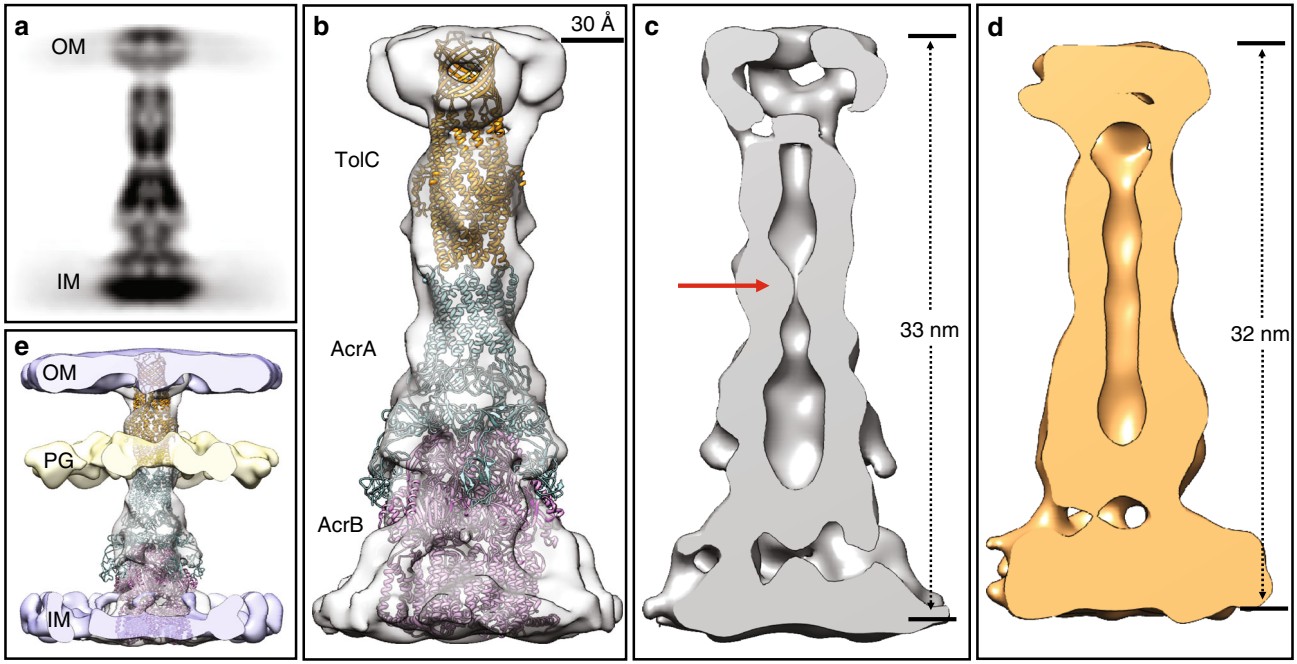

**Fig. 2** In situ Cryo-ET structures of the fully assembled AcrAB–TolC complex. **a** The side-view projection of the subtomogram average of the pump in presence of antibiotics. **b** Isosurface rendering of a fitted with high resolution cryo-EM model (PDB: 5V5S). **c** A slice through b showing a constriction at the boundary between AcrA and TolC (indicated by the red arrow). **d** A slice through the density map of the pump in presence of AcrB inhibitor (MBX3132). **e** Isosurface rendering of b overlaid with density map of the cell envelope

contrast, the structures determined from purified samples are always in an open state in the presence of antibiotic or inhibitor[19]. The significant difference between in situ and in vitro structures suggests that the OM-PG-IM envelope structure in Gram-negative bacteria and the potential between the two sides of the inner membrane may be essential for the regulation of drug efflux by keeping the conformational changes of TolC and AcrA coupled with the substrate binding of AcrB. In the cellular environment, AcrA has its N-terminal anchored in the inner membrane and its α-hairpin contacting PG, communicating between AcrB and TolC to regulate the closing and opening of the pump. Such association is disrupted during the purification, resulting in the constantly open AcrAB–TolC pump. In addition, the observation of AcrAB subcomplex suggests a critical role of PG in the assembly of the pump, which is not preserved in the purified system. With this insight, we propose that interfering with the interactions of AcrA with the PG or AcrB may interrupt the assembly process and block the function of the tripartite efflux pumps, suggesting an approach to therapeutics targeting assembly.

Our results provide the structure of the AcrAB–TolC pump and its intermediate assembly state in the native cell membrane environment. This shows the potential of in situ membrane protein structure determination with Cryo-ET. While single-particle analysis has shown great success in solving detailed protein structures, significant efforts are still needed for membrane protein purification, and the resulting structures may not truly represent their native state. The recent developments in cryo-ET make it possible to determine 10–20 Å resolution structures of membrane-embedded molecular machines[25,26], and resolve their compositional and conformational variability in the native environment.

## Methods

**Plasmid construction and protein expression**. Plasmid pAcBH which carries the *acrAB* locus and coexpresses AcrA and His-tagged AcrB was a gift from Dr. Akihito Yamaguchi (Osaka University, Suita, Japan)[27]. The *tolC* gene was first amplified using primers TolCinf_F: 5′-AAGGAGATATACATATGAAGAAATTG

CTCCCCATTCTTATCGGCC-3′ and TolCFLAGXhoI_R: 5′-GAGCTCGAGTCA CTTATCGTCGTCATCCTTGTAATCGTTACGGAAAGGGTTATGACCGTTAC TGGT-3′, and then was amplified again using TolCinf_F and TolCFLAG_inf_R: 5′-TTGAGATCTGCCCATATGTCACTTATCGTCGTCATCCTTGTAATCGTTA CG-3′. The resulting DNA fragment of *tolC-FLAG* was cloned into the pRSFDuet-1 plasmid using the In-Fusion cloning method, yielding pRSFDuet-*tolC*. *E. coli* BL21 (DE3) cells (Invitrogen) were co-transformed with plasmids pAcBH and pRSFDuet-*tolC* to overexpress AcrAB–TolC pump. Cells were cultured in 2xYT medium with 100 μg/ml ampicillin and 50 μg/ml kanamycin at 37 °C until an OD$_{600}$ of 0.8 was reached and then induced by addition of 0.1 mM isopropyl 1-thio-β-D-galactopyranoside (IPTG) at 20 °C overnight. Protein expression was examined by Coomassie blue staining and western blotting analysis.

**Minimum inhibitory concentration**. Minimum inhibitory concentration (MIC) of puromycin was measured by the twofold dilution method as described previously with minor modifications[28]. Briefly, exponentially growing cultures (OD$_{600}$ of 0.8) were inoculated at a density of $10^4$ cells per ml into LB medium containing appropriate antibiotics in the presence of twofold increasing concentrations of puromycin. Cell growth was determined visually after incubation at 37 °C for 20 h.

**In vivo crosslinking and LC/MS-MS analysis**. *E. coli* strain C43 (DE3) delta *acrAB* was co-transformed with plasmid pET20b co-expressing AcrA S273C and AcrB S258C[19] and pRSF-duet co-expressing AcrZ and TolC. The S-> C point mutations form a stabilizing disulfide bridge between AcrA and AcrB. Cells were grown in 2xYT medium with 50 μg/ml carbenicillin and 50 μg/ml kanamycin at 37 °C to OD$_{600}$ of 0.5 and then induced with 1 mM IPTG. After 2 h, cells were harvested by spinning at 4000×*g* for 5 min, then resuspended in phosphate buffered saline (PBS) supplemented with 0.2% wt/vol glucose. In vivo crosslinking of proteins to the peptidoglycan with the bifunctional 3,3′-dithiobis (sulfosuccinimidyl proprionate) (DTSSP) and isolation of the peptidoglycan with sodium dodecylsulfate followed the protocol of Li and Howard[29], with modifications. A control sample was also prepared without crosslinking. The isolated peptidoglycan from the samples were washed three times with PBS to remove traces of sodium dodecylsulfate, then incubated in PBS with 2 mg/ml lysozyme for 30 min at 37 °C to digest the peptidoglycan. The samples were centrifuged at 12,000×*g* for 10 min and the supernatants loaded onto a 4–12% gradient denaturing PAGE gel without reducing agent and stained with Coomassie brilliant blue G. A band at roughly 80 kDa that was not present in the control was excised from the gel, treated with DTT to reduce the DTSSP and then digested with chymotrypsin and analyzed by LC/MS-MS by the University of Cambridge Proteomics Facility. TolC was identified with an emPAI score of 11.1 and AcrA with a score of 10.2. Controls with bovine serum albumin (BSA) were also analyzed from the same gel, selecting a band that migrated as a dimer. The crosslinked BSA sample identified 126 peptides that had reacted with the DTSSP, while the control showed 4 false positives. The location of

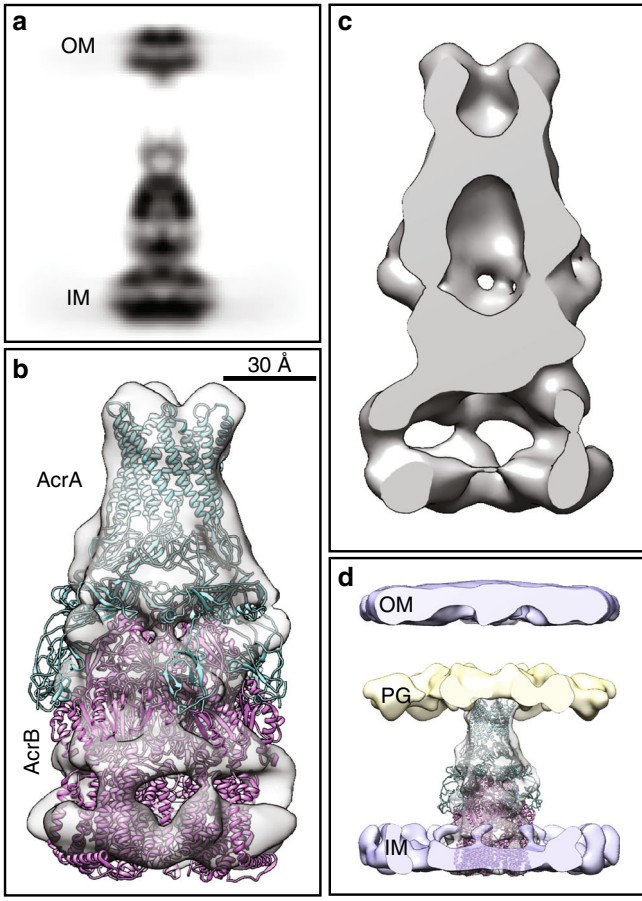

**Fig. 3** In situ Cryo-ET structure of the AcrAB subcomplex. **a** The side-view projection of the subtomogram average of the AcrAB subcomplex in presence of antibiotics. **b** Isosurface rendering of a fitted with the cryo-EM single-particle model (PDB: 5V5S). **c** A slice through the density map of **b**. **d** Isosurface rendering of **b** overlaid with density map of the cell envelope

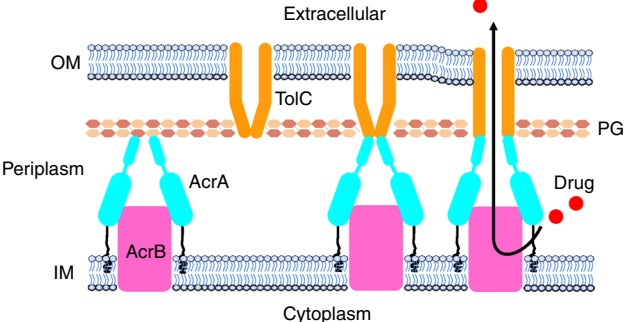

**Fig. 4** Proposed in vivo assembly and functioning mechanism for multidrug efflux pump AcrAB–TolC. First, AcrB associates with AcrA to form the bipartite complex AcrAB. Next, AcrA changes its conformation to recruit TolC. Once TolC binds with the AcrAB bipartite complex, the fully assembled tripartite pump remains in the resting state. When AcrB encounters a drug molecule, the pump adopts an open conformation accompanied with a contraction along the long axis and the substrate is expelled through the channel and out of the cell

the peptide fragments with mass corresponding to reduced DTSSP were mapped onto the crystal structures of TolC and AcrA and correspond to the equatorial domain and helical hairpin of TolC and the helical hairpin, lipoyl domain and membrane proximal domain of AcrA (see Supplementary Figs. 9, 10).

**Cryo-ET sample preparation.** *E. coli* cells were harvested and washed by PBS buffer, then resuspended to an $OD_{600}$ of 1.0. Cultures were mixed with puromycin (600 µg/ml) or MBX3132 (1.4 µg/ml) and incubated at 37 °C for half an hour. Subsequently, cells were mixed with a solution of 10 nm BSA fiducial gold (Aurion) immediately before freezing in a 1:3 cell solution to BSA gold fiducial. A single 3 µl droplet of the sample was applied to the freshly glow-discharged, continuous floating carbon film covered grids (Quantifoil Au R3.5/1, 200 mesh) and plunge frozen using a Vitrobot Mark IV (FEI). Grids were stored in liquid nitrogen until required for data collection.

**Cryo-ET data collection and 3D reconstructions.** The frozen-hydrated samples were imaged on a JEOL 3200FSC (JEOL) operated at 300 kV using a K2 Summit direct electron detector camera (GATAN), with a magnification of ×10,000 for antibiotic treated cells and ×2,000 for MBX3132 treated cells. The pixel size is calibrated to be 3.366 Å and 2.75 Å, respectively. SerialEM[30] was used to collect low-dose, single-axis tilt series at −3 to −6 µm defocus range with an average cumulative dose of ~76 e−/Å² distributed over 33 images and covering an angular range of −50° to +50°, with an angular increment of 3°. Tilted images were automatically aligned and reconstructed using EMAN2 software[31,32]. In total, more than 70 tomograms were generated to provide a sufficient selection for further processing. Supplementary Table 2 summarizes the Cryo-ET data analysis and validation statistics.

**Subtomogram averaging and correspondence analysis.** Twenty-five high SNR particles were used for initial model generation. A two-step approach was used to build the initial model. First 5 iterations of the EMAN2 initial model generation routine were performed imposing C1 symmetry. After aligning the result to the symmetry axis, we performed 5 more iterations with C3 symmetry, and used the resulting map as the initial model for subtomogram refinement. Subtomogram averaging was then performed using 1321 particles from 9 tomograms while applying C3 symmetry. This map was then used as the initial model for the following subtomogram refinement. To focus on the protein while preserving information from the membrane for improved alignment, a mask with values ranging from 0.5 to 1 around the pump and 0 to 0.5 covering a larger cylinder (~270 Å) was used for the iterative subtomogram refinement (see Supplementary Fig. 11). The refinement is performed in a gold-standard fashion with all particles split into two subsets and resolution is measured by the Fourier shell correlation of the density maps from the two subsets. After 3 iteration of subtomogram refinement, a 19 Å resolution averaged structure was achieved.

As described in the main text, the TolC region in the averaged map has lower occupancy. For further classification, we used a soft spherical mask covering the TolC region. Two initial models were generated from the averaged map obtained from the last step, one with 0.5× intensity value while the other has 1.5× intensity value inside the mask. The two maps are identical outside the mask. The two maps were used to seed a focused classification under the spherical mask for 5 iterations, resulting in two classes of particles and their corresponding density maps (see Supplementary Fig. 5).

The two classes of particles are then subject to 3 iterations of subtomogram refinement separately, using their corresponding initial model and the same mask. The subtomogram refinement is followed by 3 iterations of sub-tilt refinement, which produced the two structures shown in the figures.

To solve the structure of the pump with inhibitor, 678 particles from 10 tomograms are used for subtomogram averaging. The same initial model from the previous dataset was filtered to 50 Å and used as the initial model for the subtomogram averaging. The 21 Å structure shown in the figure was achieved after 3 iterations of subtomogram refinement.

Visualization and model docking is performed in UCSF Chimera[33] and the built-in fit in map tool.

**Reporting summary.** Further information on research design is available in the Nature Research Reporting Summary linked to this article.

## Data availability

Data supporting the findings of this manuscript are available from the corresponding authors upon reasonable request. A reporting summary for this article is available as a Supplementary Information file. The source data underlying Supplementary Figs 1a is provided as a Source Data file. Raw data for the chemical cross-links (source data for Supplementary Fig. 9 and Supplementary Fig. 10) is available via Zenodo data repository with a DOI (https://doi.org/10.5281/zenodo.2656660). The cryo-ET structure of the AcrAB subcomplex, AcrAB–TolC close state and AcrAB–TolC open state were deposited in the EMDB under ID codes EMD-0531, EMD-0532 and EMD-0533, respectively. Raw cryo-ET data are available from the corresponding author upon reasonable request.

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

## Acknowledgements

This work was supported by the Welch Foundation (Q-1967–20180324), BCM BMB department seed funds, NIH R01GM080139 and P01GM121203. B.F.L. and D.D. are supported by an ERC Advanced Award. We thank X. Yu, T. Huo and H. Wu for helpful suggestions on sample preparation, MIC assay and initial model drawing; S. Raveendran for data backup. We thank Tim Opperman and colleagues for the kind gift of the AcrB inhibitor.

## Author contributions

Z.W. designed the experiments; S.L. developed computational methods; X.S. and Z.Y. performed the experiments; B.F.L. performed the in vivo crosslinking experiments. M.C. performed computational analyses; X.S., I.F., J.J., and H.V. screened samples and collected data; X.S., M.C., H.W., J.B., and Z.W. analyzed data; X.S., M.C., and Z.W. wrote the manuscript; D.D., B.F.L., and S.L. reviewed and edited the manuscript.
