## [Peer Review File · Nature Communications]

Reviewers' Comments:

Reviewer #1:

Remarks to the Author:

REVIEW

In this manuscript by Shi et al., the authors have employed cryo-electron tomography (CET) to visualize the AcrAB-TolC multidrug efflux pump in situ in E-coli. Using the subtomogram averaging, the authors have determined a decent (impressive by CET standards) resolution structure of the pump in both open and closed state. By subjecting the E-coli to antibiotic or a known inhibitor of AcrA, the authors were able to subject the pump to closed/open conformation and determined its both closed and open conformation. By performing statistics on the multiple occurrences of the pump in the tomograms (sometimes via classification), the authors were also able to suggest that a> the pump primarily exists in the closed state and is open only transiently to flux out the antibiotic since almost all the occurrences of the pump in the tomograms is in the closed state. b> The final assembly of the AcrAB-TolC occurs between a preformed AcrAB (exists on the inner membrane) and TolC (exists on the outer membrane).

The experiments and data analysis are expertly performed. The number of tomograms and the number of particles extracted for subtomogram is quite impressive considering these experiments and data analysis require significant efforts and skills. The manuscript is well-written and figures are well-presented. I recommend the publication of this manuscript, but I have some comments that would need to be addressed:

- In some instances, the manuscript seems quite firm in its statements about the mechanism of the pump's action, and some of my comments are related to this.

- o Statements about the mechanism of the assembly of AcrAB seems quite speculative.

'AcrB located in the inner membrane associates with AcrA to form bipartite complex AcrAB.' Is it known that these complexes form in the membrane or they exist in a pre-formed state in the membrane?

- o In Extended Fig.4, I failed to see the points about AcrA conformation changes 'The reorganization of the AcrA hexamer might be critical step for the recruitment of TolC to form the tripartite pump complex.'. Can the authors present this more clearly?

- Authors have a figure showing the local resolution in the supplementary/extended figure. In the same extended figure, can they add local resolution maps for the map shown in Fig. 2c-2d. This addition would help the readers see the local resolution around the area where the pump constricts.

- By boxing out small subtomograms of the interface at which the AcrAB interacts with PG, would it be possible to obtain higher resolution details of the interaction site between AcrAB and PG? Because I am guessing that the interaction site may be one of the most effective drug targeting sites. Authors need not do this analysis, but it would be good to get their insights about this.

- The authors could have used a higher cumulative dose of up to $\sim 100 \text{ e}^-/\text{\AA}^2$ instead of $\sim 76 \text{ e}^-/\text{\AA}^2$. Was there any deliberate reason for the lower dose?

The following comments and questions need not be incorporated into the manuscript. These are my general curiosities. However, the authors may choose to include some of these points in the manuscript as well.

- How does the behavior/response of the E-coil change to the antibiotic in the presence of AcrB inhibitor (MBX3132)? Does it become hyper-resistant? I am guessing the cells die quite quickly due to unregulated exposure through this pump.

- How and when does the pump disassemble? Through the same pathway of assembly i.e. AcrAB

and TolC disassociate?

- To respond to increased antibiotic stress, does the E-coli recruit more pumps or do the existing pump open more frequently? I am guessing it is the latter since you never observed the open pumps in situ.

Reviewer #2:

Remarks to the Author:

The data and results in this paper are novel and provide a nice incremental step forward in our understanding of the AcrAB-TolC multi-drug efflux pump, but I have some concerns about the final model presented.

Merits of this work:

1. Structure of AcrAB-TolC resolved in vivo for the first time, including in two different states
2. An intermediate state without TolC identified, which reveals insight into the assembly process
3. Peptidoglycan layer is shown to contact both AcrA and TolC rather than the previously proposed equatorial domain of TolC

Questions:

1. How were the particles picked from the tomograms? By eye? EMAN2's machine learning?
2. Were the images CTF-corrected? If so, how?
3. In Figure 3a (the AcrAB complex), there is a strong density in the outer membrane region. If TolC is absent, why would the outer membrane be positioned so consistently? Note the density of the OM appears stronger in the AcrAB complex than the AcrAB-TolC complex. How does this relate to the claim that the presence of the AcrAB-TolC complex pinches the periplasm slightly and holds it at a constant width. How would this happen in the absence of TolC? I wonder if this is an artifact from mis-classification of particles.
4. Why don't the authors perform the same focused classifications for the open-state as they did for the closed state? Does the AcrB inhibitor promote the assembly of the full complex?
5. Local resolution analysis for the open-state structure is missing.
6. Even though the opening of AcrA is clear in the open-state structure, the rest of the complex looks similar between the two states. Why not produce an open-state average with the same unbiased procedure used for the closed state just to exclude the possibility of model bias?
9. When determining the open-state structure model, did the authors apply any symmetry?
10. The cell shown in Fig 1a appears lysed and dying (condensed material, internal membranes). That probably doesn't affect the efflux pumps, but it should be noted in the text and figure legend.
11. In line 62 it is claimed that the pump keeps the periplasm width consistent, but that is actually normal – perhaps the authors don't realize that because all their cells were lysed and/or dying like the cell in Fig. 1a.
12. Did the authors find isolated TolC complexes? Their model predicts they should exist.
13. Line 165 claim that cryo-ET can deliver "high resolution structures" of membrane-embedded molecular machines is an overstatement. Most people think "high resolution" means "near-atomic." There is no such example yet, but some are close - the authors should reference these to clarify what they mean.

My main concern, though, is that the model in Fig. 4 does not match the data in two respects: (1) the alpha-hairpin domain of AcrA is actually more closed in the TolC-bound conformation than the TolC-unbound, and (2) the constriction in the channel is actually found at the level of this alpha-hairpin domain, NOT in TolC – their model shows TolC closed, but their data says it is open, at least whenever bound to AcrAB. Some re-thinking is needed.

Reviewer #3:

Remarks to the Author:

The article "In situ structure of the multidrug efflux pump AcrAB-TolC" by Shi et al details the differences observed in structure when comparing this study (tomography on whole cells) with previous efforts (isolated complexes, "in-vitro" structural methods).

I am enthusiastic about the possibilities and usage of tomography, and have been impressed by peer studies on flagellar assembly, nucleoid arrangement, type six secretion systems and chemotaxis to name but a few. However, this particular manuscript is largely hypothesis generation, with a deficit in hypothesis testing/validation and as such I find it too limited to be recommended for publication in Nature Communications.

(For comparative purposes, a recent in-situ study with more depth can be seen in this months EMBO J - <http://emboj.embopress.org/content/early/2019/03/15/embj.2018100886>)

In particular:

Abstract: The final statement "...provides the structural basis for design of efflux pump inhibitors". This does not, not at this resolution. I find this to be an over-reaching conclusion.

Line 68: I would like a better description of use/choice of C3 averaging. The AcrAB-TolC machinery is well known to observe several structural states in different monomers of the trimer (Seeger et al. Science 2006;1295- or Murakami et al Nature 2006;443), and I worry that the choice for 3 state heterogenous trimer vs homogenous trimer is less well-described here than it was in the eLife paper by the authors on the isolated pump.

Line 90: In this validation experiment, did you test the MIC of these cells for a particular antibiotic to see it change upon MBX3132 addition?

In terms of my opinion that the paper is dominated by hypothesis generation rather than validation, I think this apparent when looking at the language on lines 121 (might), 125 (possibly), 127 (or), 133 (we suggest), 136 (may permit), 147 (suggests that), 148 (may be) etc. This study needs more validation by secondary techniques/experiments.

Lines 151-153: The argument that the association between pump components is disrupted by purification, is speculation without extra experimentation.

Line 165: "High resolution" here is misused; the study is good for tomography, but would not be best described as "high resolution".

General: Did the authors try a tomogram of cells expressing a DARPIN (instead of small molecule) to block pump function? This could be a worthwhile addition.

Point-by-point response

Reviewer #1

REVIEW

In this manuscript by Shi et al., the authors have employed cryo-electron tomography (CET) to visualize the AcrAB-TolC multidrug efflux pump in situ in E-coli. Using the subtomogram averaging, the authors have determined a decent (impressive by CET standards) resolution structure of the pump in both open and closed state. By subjecting the E-coli to antibiotic or a known inhibitor of AcrA, the authors were able to subject the pump to closed/open conformation and determined its both closed and open conformation. By performing statistics on the multiple occurrences of the pump in the tomograms (sometimes via classification), the authors were also able to suggest that a> the pump primarily exists in the closed state and is open only transiently to flux out the antibiotic since almost all the occurrences of the pump in the tomograms is in the closed state. b> The final assembly of the AcrAB-TolC occurs between a preformed AcrAB (exists on the inner membrane) and TolC (exists on the outer membrane).

The experiments and data analysis are expertly performed. The number of tomograms and the number of particles extracted for subtomogram is quite impressive considering these experiments and data analysis require significant efforts and skills. The manuscript is well-written and figures are well-presented. I recommend the publication of this manuscript, but I have some comments that would need to be addressed:

- In some instances, the manuscript seems quite firm in its statements about the mechanism of the pump's action, and some of my comments are related to this.

- o Statements about the mechanism of the assembly of AcrAB seems quite speculative.

‘AcrB located in the inner membrane associates with AcrA to form bipartite complex AcrAB.’ Is it known that these complexes form in the membrane or they exist in a pre-formed state in the membrane?

Our reply: While AcrB is an integral inner membrane protein, AcrA is a periplasmic protein with its N-terminal anchored in the inner membrane. A cross-linking study showed that AcrA/B associates to form a complex even in the absence of antibiotics or TolC in intact cells (Zgurskaya, H. I. & Nikaido, J.Bacteriol, 2000). We have clarified this in the revised text.

- o In Extended Fig.4, I failed to see the points about AcrA conformation changes ‘The reorganization of the AcrA hexamer might be critical step for the recruitment of TolC to form the tripartite pump complex.’. Can the authors present this more clearly?

Our reply: Our earlier work from single particle analysis of the AcrAB-TolC assembly showed that the alpha-helical barrel portion of AcrA repacks in the presence of either inhibitor or antibiotics (Wang *et al.*, eLife, 2017). We have changed the wording to read: “The helical

hairpin region of AcrA repacks to form an alpha-helical barrel in the transition from the apo to the ligand-bound states of the AcrAB-TolC assembly, and the reorganisation of the AcrA hexamer is likely to be a critical step for the opening of the TolC channel to form the active tripartite pump complex.”

- Authors have a figure showing the local resolution in the supplementary/extended figure. In the same extended figure, can they add local resolution maps for the map shown in Fig. 2c-2d. This addition would help the readers see the local resolution around the area where the pump constricts.

Our reply: We have added the local resolution representations for the maps shown in Fig. 2c-2d in Supplementary Figure 3 and Supplementary Figure 4.

- By boxing out small subtomograms of the interface at which the AcrAB interacts with PG, would it be possible to obtain higher resolution details of the interaction site between AcrAB and PG? Because I am guessing that the interaction site may be one of the most effective drug targeting sites. Authors need not do this analysis, but it would be good to get their insights about this.

Our reply: We appreciate the suggestion from the reviewer. The feature size of this region is already a challenge for *in situ* subtomogram averaging. With the current data, using a smaller size for alignment does not work well. We are currently working on a massive data collection with optimized imaging condition and development of new focused alignment techniques to improve the resolution of the particular part of structure. The interaction between AcrAB and PG is certainly one of the interests of our further research.

- The authors could have used a higher cumulative dose of up to $\sim 100 \text{ e}^-/\text{\AA}^2$ instead of $\sim 76 \text{ e}^-/\text{\AA}^2$. Was there any deliberate reason for the lower dose?

Our reply: From single particle studies we already know that high resolution features begin to suffer radiation damage at doses as low as $10\text{-}12 \text{ e}^-/\text{\AA}^2$. In cellular tomography it is conventional to use doses up to the point at which bubbling occurs (indicating very severe radiation damage). For our cells this is at $\sim 80 \text{ e}^-/\text{\AA}^2$. However, if we are averaging tomograms, and hope to achieve subnanometer resolution in the future, we will need to restrict the total dose even further. It is very likely that the high total dose in the current study is limiting our resolution.

The following comments and questions need not be incorporated into the manuscript. These are my general curiosities. However, the authors may choose to include some of these points in the manuscript as well.

- How does the behavior/response of the E-coil change to the antibiotic in the presence of AcrB inhibitor (MBX3132)? Does it become hyper-resistant? I am guessing the cells die quite quickly due to unregulated exposure through this pump.

Our reply: *E. coli* cells do become hyper-resistant to puromycin in the presence of MBX3132. We have added data related to this point in supplementary table 1.

• How and when does the pump disassemble? Through the same pathway of assembly i.e. AcrAB and TolC disassociate?

Our reply: In principle, these processes are binding events and therefore reversible. Since we are only observing assemblies frozen at a time point in our cryo-ET results, it is difficult to differentiate between the assembly and disassembly processes. Additional biochemical and cryo-ET experiments may be needed to distinguish the assembly/disassembly processes and explore if they are linked to an irreversible process, such as energy transduction.

• To respond to increased antibiotic stress, does the E-coli recruit more pumps or do the existing pump open more frequently? I am guessing it is the latter since you never observed the open pumps in situ.

Our reply: It seems possible that increased antibiotic stress will result in both greater recruitment of pumps and more frequent opening of both the pre-existing and newly formed pumps.

Reviewer #2

The data and results in this paper are novel and provide a nice incremental step forward in our understanding of the AcrAB-TolC multi-drug efflux pump, but I have some concerns about the final model presented.

Merits of this work:

1. Structure of AcrAB-TolC resolved in vivo for the first time, including in two different states
2. An intermediate state without TolC identified, which reveals insight into the assembly process
3. Peptidoglycan layer is shown to contact both AcrA and TolC rather than the previously proposed equatorial domain of TolC

Questions:

1. How were the particles picked from the tomograms? By eye? EMAN2's machine learning?

Our reply: The pump in this dataset is highly abundant and featureful, making it easy to select either by eye or machine. In this case, the project was conducted during the development of the tomogram workflow interface in EMAN2, and the particles were selected manually to ensure every part of the program is working properly. Machine learning picking was also tested and produced similar results.

2. Were the images CTF-corrected? If so, how?

Our reply: We apologize for missing this important technical detail. The particles are CTF corrected using a per-particle-per-tilt method in the EMAN2 tomogram workflow. The details of

the method are available in the accessible manuscript (arXiv:1902.03978), which is currently under review.

3. In Figure 3a (the AcrAB complex), there is a strong density in the outer membrane region. If TolC is absent, why would the outer membrane be positioned so consistently? Note the density of the OM appears stronger in the AcrAB complex than the AcrAB-TolC complex. How does this relate to the claim that the presence of the AcrAB-TolC complex pinches the periplasm slightly and holds it at a constant width. How would this happen in the absence of TolC? I wonder if this is an artifact from mis-classification of particles.

Our reply: The strong density in the outer membrane is actually a patch of the outer membrane with no protein. It appears due to the cylindrical masking based on the position and orientation of the AcrAB complex and is not related to TolC. We have added an additional figure showing how the complexes with and without TolC are distributed in one of the original cells (see Supplementary Figure 6). The complete complexes are regularly distributed, and each of them acts as a local anchor for the inner/outer membrane spacing, making the distance more self-consistent. The complexes without TolC are not expected to contribute to this effect.

4. Why don't the authors perform the same focused classifications for the open-state as they did for the closed state? Does the AcrB inhibitor promote the assembly of the full complex?

Our reply: Unsupervised classification requires a larger population of particles than we currently have for this state. We are aiming to collect much larger datasets for multiple states which will allow us to address statistically in the future whether the inhibitor promotes full complex assembly.

5. Local resolution analysis for the open-state structure is missing.

Our reply: We thank the reviewer for the suggestion. A local resolution map for the open-state structure is now shown in Supplementary Figure 4.

6. Even though the opening of AcrA is clear in the open-state structure, the rest of the complex looks similar between the two states. Why not produce an open-state average with the same unbiased procedure used for the closed state just to exclude the possibility of model bias?

Our reply: We solved the open structure in two approaches: i) using the same initial model as the closed state, and ii) from scratch using unbiased methods. The first approach we feel is actually stronger evidence for the structural change because it sets a bias for the closed that is nonetheless overcome by the experimental data. When the structure is solved with different starting models the possibility that noise bias from the initial model induced some of the changes exists. If we start from the same starting models for the inhibitor (open) and antibiotic (closed) and achieve different results, then we are more confident that the features we observe are derived from the data rather than the model. The figure below shows the structure solved using the second approach way; it is very similar to the result using the first approach.

Fig.1 Open-state average with the same unbiased procedure used for the closed state.

9. When determining the open-state structure model, did the authors apply any symmetry?

Our reply: We applied C3 symmetry in the refinement of both open and closed state structures as discussed in the methods.

10. The cell shown in Fig 1a appears lysed and dying (condensed material, internal membranes). That probably doesn't affect the efflux pumps, but it should be noted in the text and figure legend.

Our reply: These are actually overexpressing less AcrAB and TolC than the cells we use for purification for single particle studies. They are clearly still sufficiently healthy for replication/growth. So, while we are aware of the large amount of condensed material in the cells, and they are certainly less healthy than they would be in the absence of overexpression, they are still alive and viable, and are very resistant to antibiotics. We have added a brief mention of this to the text and figure legend.

11. In line 62 it is claimed that the pump keeps the periplasm width consistent, but that is actually normal – perhaps the authors don't realize that because all their cells were lysed and/or dying like the cell in Fig. 1a.

Our reply: We believe the cell in Fig 1a is healthy (healthy enough for reproduction), and it is not lysed. The impression that the cells are lysed or dying might be from looking at the upper edge of the membrane when it becomes poorly defined, but this is an artifact of the tomography process, due to missing information in parts of the tomogram, and is not observed in individual tilt images. That is, the cell is intact, not lysed. Supplementary Figure 2 shows a wild type cell without overexpression, and the distance between inner and outer membranes in these wild type cells is not constant in the absence of the complexes.

12. Did the authors find isolated TolC complexes? Their model predicts they should exist.

Our reply: It is difficult to identify smaller objects such as isolated TolC unambiguously. We would point out that our AcrAB complexes without TolC were identified as a subclass of putative complex particles. There are some densities present which could represent TolC alone, but it could also be other outer membrane proteins, and we felt we could not do this unambiguously without a labelling approach.

13. Line 165 claim that cryo-ET can deliver “high resolution structures” of membrane-embedded molecular machines is an overstatement. Most people think “high resolution” means “near-atomic.” There is no such example yet, but some are close - the authors should reference these to clarify what they mean.

Our reply: We have modified the statement to read “The recent developments in cryo-ET make it possible to determine 10-20 Å resolution structures of membrane embedded molecular machines” and added two corresponding references to clarify what we mean. We believe we will be able to achieve subnanometer resolutions with better optimizations of imaging conditions in the near future.

My main concern, though, is that the model in Fig. 4 does not match the data in two respects: (1) the alpha-hairpin domain of AcrA is actually more closed in the TolC-bound conformation than the TolC-unbound, and (2) the constriction in the channel is actually found at the level of this alpha-hairpin domain, NOT in TolC – their model shows TolC closed, but their data says it is open, at least whenever bound to AcrAB. Some re-thinking is needed.

Our reply: (1) We agree that we are somewhat constrained by our resolution. Taking these comments into account we have modified our model and regenerated Fig. 4 according to the data. The alpha-hairpin domain of AcrA in the full pump is more closed than that in the AcrAB subcomplex. (2) Due to the resolution limitation in our data, we cannot rule out TolC closure in the constriction position. We have added the following line in the discussion of the model to address this: “Notably, the *in situ* fully assembled “closed state” pump showed a constriction in between TolC and AcrA”.

Reviewer #3

The article “In situ structure of the multidrug efflux pump AcrAB-TolC” by Shi et al details the differences observed in structure when comparing this study (tomography on whole cells) with previous efforts (isolated complexes, “in-vitro” structural methods).

I am enthusiastic about the possibilities and usage of tomography, and have been impressed by peer studies on flagellar assembly, nucleoid arrangement, type six secretion systems and chemotaxis to name but a few. However, this particular manuscript is largely hypothesis generation, with a deficit in hypothesis testing/validation and as such I find it too limited to be recommended for publication in Nature Communications.

(For comparative purposes, a recent in-situ study with more depth can be seen in this month's EMBO J - <http://emboj.embopress.org/content/early/2019/03/15/embj.2018100886>)

Our reply: While we appreciate the reviewer's point of view and have now included the noted reference and other publications in our revised manuscript, we disagree with the conclusion about the relevance of our work. We also point out that despite not using any of the advanced technologies available in that other study (FIB milling and phase plates), our study achieved significantly better resolution (15 Å) than that other mentioned study (20 Å; 1.7 MDa; C5), and our complex is much smaller (~760 kDa) and has lower symmetry (C3), making this system even more challenging. We believe that in addition to the technological achievement, that observation of the constraints on membrane separation, distribution of complexes and other features are more than mere "hypothesis generation" and combined these factors merit publication in Nature Communications. We believe the manuscript will be of interest to both workers in this field as well as practitioners of Cryo-ET, and cell biologists in general.

In particular:

Abstract: The final statement "...provides the structural basis for design of efflux pump inhibitors". This does not, not at this resolution. I find this to be an over-reaching conclusion.

Our reply: We have changed the final statement to read "...provides structural insights for design of efflux pump inhibitors".

Line 68: I would like a better description of use/choice of C3 averaging. The AcrAB-TolC machinery is well known to observe several structural states in different monomers of the trimer (Seeger et al. Science 2006;1295- or Murakami et al Nature 2006;443), and I worry that the choice for 3 state heterogenous trimer vs homogenous trimer is less well-described here than it was in the eLife paper by the authors on the isolated pump.

Our reply: Whether to apply C3 symmetry is certainly one of our concerns when processing this dataset. No biological structure has absolute symmetry to atomic resolution, so the question is at what resolution it becomes useful to discard the symmetry assumption. Based on our earlier published structure, at the 15 Å resolution we have achieved, symmetry breaking through non-equivalence of the AcrB protomers with antibiotic binding will not yet be detectable. Moreover, our analysis of the complex in the presence of the inhibitor shows that it has C3 symmetry (Wang *et al.*, eLife, 2017). As we move to higher resolutions we will test whether symmetry should be applied, but in the present study it is clearly justified.

Line 90: In this validation experiment, did you test the MIC of these cells for a particular antibiotic to see it change upon MBX3132 addition?

Our reply: We tested the MIC of these cells for a particular antibiotic and found that they became hyper-resistant to puromycin upon MBX3132 addition. We have added these data in supplementary table 1.

In terms of my opinion that the paper is dominated by hypothesis generation rather than validation, I think this apparent when looking at the language on lines 121 (might), 125 (possibly), 127 (or), 133 (we suggest), 136 (may permit), 147 (suggests that), 148 (may be) etc. This study needs more validation by secondary techniques/experiments.

Our reply: It is true that we used weaker language in places than we might have, but we disagree that this alone means all we are doing is hypothesis generation. Indeed every scientific conclusion is really just hypothesis generation from this perspective. Many of our points are validated by external data, and we disagree with the reviewer's assessment.

Lines 151-153: The argument that the association between pump components is disrupted by purification, is speculation without extra experimentation.

Our reply: Since AcrAB is associated with the inner membrane and TolC is associated with the outer membrane, clearly the full complex will not survive typical purification. This is not speculation, this is logically obvious. By looking at the intact cell, clearly we are able to see the complex as it forms in the cell, whereas recombining the component *in vitro* could potentially induce artifacts, particularly since the forces applied by the two associated membranes would be absent.

Line 165: "High resolution" here is misused; the study is good for tomography, but would not be best described as "high resolution".

Our reply: The 2nd reviewer also noted this and we agree. We have rephrased this.

General: Did the authors try a tomogram of cells expressing a DARPIN (instead of small molecule) to block pump function? This could be a worthwhile addition.

Our reply: While this is a good suggestion, we are not clear how DARPIN will localize to the periplasm where it would bind the AcrB periplasmic headpiece. We are continuing this work with a wider range of conditions, with larger data collections and optimized imaging conditions for future work, but we believe the present study is of sufficient importance for publication on its own. Indeed adding many additional conditions would likely be of more interest to specialists rather than a general audience.

Reviewers' Comments:

Reviewer #1:

Remarks to the Author:

The authors have answered all my queries with good clarifications. This final product can be published.

Reviewer #2:

Remarks to the Author:

All of my concerns were addressed satisfactorily. Concerning Reviewer #3's comments and the authors' responses, I agree with the authors that the manuscript will be of interest to both efflux-pump researchers and practitioners of cryo-ET. I agree with Reviewer #3 that the abstract should not claim this work establishes a basis for drug design. I think the author's revised sentence "provides structural insights for design of efflux pump inhibitors" should be even further clarified, for instance to something like "suggests new domains to target with efflux pump inhibitors."

Point-by-point response

Reviewer #1

The authors have answered all my queries with good clarifications. This final product can be published.

Reviewer #2

All of my concerns were addressed satisfactorily. Concerning Reviewer #3's comments and the authors' responses, I agree with the authors that the manuscript will be of interest to both efflux-pump researchers and practitioners of cryo-ET. I agree with Reviewer #3 that the abstract should not claim this work establishes a basis for drug design. I think the author's revised sentence "provides structural insights for design of efflux pump inhibitors" should be even further clarified, for instance to something like "suggests new domains to target with efflux pump inhibitors."

Our reply: We have changed the final statement to read "...suggest domains to target with efflux pump inhibitors".